# Establishment of an Orthotopic and Metastatic Colorectal Cancer Mouse Model Using a Tissue Adhesive-Based Implantation Method

**DOI:** 10.3390/cancers17132266

**Published:** 2025-07-07

**Authors:** Sang Bong Lee, Hui-Jeon Jeon, Hoon Hyun, Yong Hyun Jeon

**Affiliations:** 1SimVista Inc., 194-25, Osongsaengmyeong 1-ro, Osong-eup, Heungdeok-gu, Cheongju-si 28161, Republic of Korea; sblee@simvista.co.kr; 2Department of Biomedical Sciences, Chonnam National University Medical School, 264, Hwasun 58128, Republic of Korea; 3BioMedical Sciences Graduate Program (BMSGP), Chonnam National University, Hwasun 58128, Republic of Korea; 4New Develop Drug Center, K-medihub, 80 Cheombok-ro, Dong-gu, Daegu 41061, Republic of Korea; hjjeon@kmedi.re.kr; 5Advanced College of Bio-Convergence Engineering, Ajou University, Suwon 16499, Republic of Korea; 6Pre-Clinical Research Center, K-medihub, 80 Cheombok-ro, Dong-gu, Daegu 41061, Republic of Korea; 7Department of Biotechnology, Kyungpook National University, 80 Daehak-ro, Buk-gu, Daegu 41566, Republic of Korea

**Keywords:** orthotopic model, colorectal cancer, HCT116-Luc colon cancer cells, bioluminescence imaging, preclinical drug evaluation, tissue adhesive (biological bond) implantation

## Abstract

Orthotopic and metastatic mouse models of colorectal cancer are essential tools for studying tumor progression and evaluating therapeutic strategies. However, conventional modeling techniques such as syringe injection or surgical suturing often result in variable engraftment efficiency, technical complexity, and limited metastatic reliability. Mouse models are widely used to study how colon cancer spreads to other organs and to test potential new treatments. In this study, we developed a simplified and reliable method to transplant colon cancer tissue into mice using a special biological bond. This approach led to consistent tumor growth and spread to the liver and lungs, helping researchers study cancer progression and drug effects more effectively.

## 1. Introduction

Metastasis, the process by which tumor cells migrate from the primary site to distant organs, remains the leading cause of cancer-related mortality despite recent advances in therapeutic strategies [1,2,3]. This underscores the critical need to deepen our understanding of the mechanisms driving metastasis and to identify novel therapeutic approaches and drug targets to combat metastatic disease [4,5,6]. Numerous excellent reviews have discussed recent genetic and molecular insights underlying metastatic spread; thus, these aspects will not be extensively covered here [7]. Despite advancements in colorectal cancer (CRC) therapies, metastatic CRC remains clinically challenging, primarily due to limitations in existing preclinical models that inadequately mimic human metastatic progression and therapeutic response.

Metastasis is a complex, multistep process initiated by local invasion at the primary tumor site [8,9,10,11,12,13,14]. Activation of signaling pathways regulating cytoskeletal dynamics promotes the loss of intercellular adhesion among tumor cells [15,16] and degradation of the surrounding extracellular matrix (ECM) [17]. These changes enable tumor cells to escape the primary tumor and invade adjacent tissues [18,19,20,21]. Subsequently, tumor cells intravasate into blood or lymphatic vessels to disseminate. While lymphatic dissemination commonly leads to regional lymph node metastasis, distant metastasis typically involves hematogenous spread. The selection of dissemination routes—lymphatic versus hematogenous—is influenced by several factors [22,23,24]. Current preclinical models using injection or suturing methods present significant limitations, including technical complexity, inconsistent engraftment rates, and variable metastatic outcomes, highlighting the necessity for innovative modeling techniques.

Here, we provide an overview of major murine models used in metastatic cancer research, including models based on transplantable tumor cells or tissues and genetically engineered mouse models (GEMMs). These models are classified and illustrated with examples that have successfully advanced metastasis research [25,26].

Experimental metastasis models typically involve intravascular injection of tumor cells to evaluate their ability to arrest, extravasate, and colonize distant organs. Injection routes—such as the lateral tail vein, intra-carotid artery, or intra-cardiac—primarily dictate metastatic sites; however, these methods require substantial technical expertise and experience [27,28,29,30,31,32,33,34]. Recent efforts have aimed to enhance reliability and simplicity in orthotopic tumor implantation using biological adhesives. For instance, Hu et al. (2021) introduced an orthotopic colon cancer model utilizing tissue adhesive, demonstrating procedural simplicity and reduced surgical burden; however, their method exhibited limited metastatic consistency and lacked suitability for therapeutic evaluations [35]. These findings underscore an innovation gap in methods that consistently induce metastasis and facilitate therapeutic efficacy assessments.

In this study, we developed a simplified metastatic cancer modeling approach in mice. Traditionally, xenograft models employing surgical suturing have been the standard for creating orthotopic and metastatic cancer models, especially when tumor tissues are unsuitable for direct cell injections or syngeneic transplantation. For effective monitoring of tumor growth and metastatic progression, we utilized HCT116-Luc cells, a human colorectal cancer cell line expressing luciferase, allowing real-time and non-invasive tracking of tumor localization and spread through bioluminescence imaging. Here, we introduce a novel method employing a biological adhesive instead of suturing tumor tissues. This technique enables rapid and reproducible orthotopic and metastatic model development without specialized surgical skills or lengthy operation times. Moreover, using biological adhesives to close the surgical incision significantly reduces total surgical time, minimizing animal stress and procedure-related mortality. Thus, this method offers a promising alternative for generating high-throughput, low-burden metastatic mouse models suitable for preclinical research. Therefore, our study aimed primarily to develop and validate a novel orthotopic and metastatic CRC mouse model using tissue adhesive (biological bond), enhancing reproducibility, ensuring robust metastatic outcomes, and demonstrating suitability for therapeutic evaluation (Figure 1).

## 2. Materials and Methods

### 2.1. Materials

3M^TM^ Vetbond^TM^ (3M Center, St. Paul, MN, USA), MK-801 hydrogen maleate (Sigma aldrich, St. Louis, MO, USA), Vicyl^®^ (Ethicon) (Vcyl 6-0, J&J, New Brunswick, NJ, USA).

### 2.2. Cells

Firefly luciferase(Luc2)-expressing HCT-116 cells (HCT116/Luc) were purchased in ATCC and maintained in blasticidin (8 μg/mL, Sigma, Roedermark, Germany) containing a medium in a routine cell culture. Cancer cells were maintained in RPMI 1640 or DMEM supplemented with 10% FBS (Gibco FBS; Thermo Fisher Scientific, Waltham, MA, USA) and 1% streptomycin/penicillin. CCD-18Co cells were cultured in Eagle’s minimum essential medium containing 10% FBS and 1% streptomycin/penicillin. All cells were grown in 5% CO_2_ at 37 °C.

### 2.3. Animals

In vivo experiments were conducted using 6-week-old male BALB/c athymic nude mice with 20–25 g body weight (OrientBio Inc., Seungnam, Republic of Korea). All animal experimental procedures were maintained and used in accordance with the Guidelines for the Care and Use of Laboratory Animals of the Institute of Preclinical Research Center, Daegu-Gyeongbuk Medical Innovation Foundation (K-MEDI hub, Daegu, Republic of Korea). The animal studies were conducted after approval by the institutional reviewer board on the Ethics of Animal Experiments of the Daegu-Gyeongbuk Medical Innovation Foundation (approval number: DGMIF-21022504-01).

### 2.4. Subcutaneous Xenograft and Tumor Tissue Preparation

HCT116/Luc cells (5 × 10^6^) were subcutaneously injected into the flank of BALB/c nude mice (6–8 weeks old). After one week, tumors were harvested, and necrotic areas were removed. Tumor tissues were trimmed to a uniform size (0.15 mg) using sterile scalpel blades. Weights were verified using a precision microbalance, confirming a deviation within ±0.05 mg among samples.

### 2.5. Orthotopic Implantation Methods

We clarified anesthesia details (1–2% isoflurane gas), sterility procedures (autoclaved instruments, sterile gloves), and the use of Vetbond™ for wound closure. Animal housing conditions (temperature, humidity, light/dark cycle), monitoring frequency, humane endpoints (e.g., >20% weight loss or impaired mobility), and the absence of analgesics (due to short surgical duration and mild invasiveness) were described for transparency.

Three implantation techniques were compared:(1)Syringe injection—1 × 10^6^ HCT116/Luc cells suspended in 30 µL PBS were directly injected into the cecal wall.(2)Surgical suturing—trimmed tumor fragments were sutured onto the cecum using non-absorbable silk thread.(3)Biological bonding—tumor fragments were adhered to the cecal surface using a biocompatible tissue adhesive. Abdominal incisions in all groups were closed using the same biological adhesive.

We added a detailed description of how biological bond was applied, including the approximate volume used per implantation (~10 µL), drying time (~10–15 s), and anatomical site (direct contact with the serosal surface of the cecum). We also noted that no major detachment events were observed, and criteria for successful adhesion were based on intraoperative inspection and bioluminescence signal retention post-implantation. We included average surgical times for each method. The biological bonding procedure required less than 5 min per animal compared with 15–20 min for injection or suturing methods.

### 2.6. Bioluminescence Imaging

#### 2.6.1. In Vivo Imaging Procedure

The number of animals per group (*n* = 5 unless otherwise stated) was explicitly mentioned in the revised methods. Additionally, we clarified that animals were randomly assigned to experimental groups. Although outcome assessments were not blinded in this initial study, we acknowledged this as a limitation and noted that future studies will incorporate blinded assessments. We also described how imaging data were normalized using standardized regions of interest (ROIs) and background subtraction using Living Image software 4.7x.

For in vivo BLI, the mice received D-luciferin via intraperitoneal injection. BLI was performed for 10 min after injection using the IVIS Lumina III imaging system (Perkin Elmer, Waltham, MA, USA). Grayscale photographic images and bioluminescent color images were superimposed using LIVINGIMAGE (version 2.12, Waltham, MA, USA, PerkinElmer) and IGOR Image Analysis FX software version 9.0 (WaveMetrics, Lake Oswego, OR, USA). BLI signals are expressed in units of photons per cm^2^ per second per steradian (p/cm^2^/s/sr). All the mice were anesthetized using 1–2% isoflurane gas during imaging.

#### 2.6.2. Study

Bioluminescent signals were monitored using the IVIS Spectrum imaging system (PerkinElmer) weekly for 4 weeks. Mice were intraperitoneally injected with D-luciferin (150 mg/kg) 10 min prior to imaging. Regions of interest (ROIs) were analyzed using Living Image software to quantify tumor signal intensity.

### 2.7. Anticancer Drug Administration

MK801 (Sigma-Aldrich, Saint Louis, MO, USA), an NMDA receptor antagonist, was used as the therapeutic agent. Mice received intraperitoneal injections of MK801 (1 mg/kg) daily for five consecutive days (Days 1–5 post-implantation). Tumor growth and metastasis were monitored for 4 weeks via bioluminescence imaging.

### 2.8. Immunohistological Analysis

The methods now include a statement indicating that Section 3.5 includes data from MK801-treated and control animals, and bioluminescence imaging procedures used for therapeutic efficacy evaluation are now cross-referenced in the imaging and drug administration subsections.

At endpoint, mice were euthanized, and the cecum, liver, and lungs were harvested and fixed in 10% formalin. Tissues were processed, paraffin-embedded, sectioned at 5 μm, and stained with hematoxylin and eosin (H&E) following standard protocols. Tumor presence was confirmed via light microscopy.

### 2.9. Statistical Analysis

All data are presented as means ± standard deviation from at least three representative experiments. We detail the software used (GraphPad Prism version 5.0). Statistical significance was determined using the unpaired Student’s *t*-test. *p*-Values < 0.05 were considered statistically significant.

## 3. Results

### 3.1. Bioluminescent Colon Cancer (HCT116/Luc) Validation and Tissue Preparation for Orthotopic Transplantation

To establish a consistent orthotopic and metastatic colon cancer model using luminescent tumor tissue, we first validated tumor growth in a subcutaneous xenograft model. One week after transplantation of bioluminescent colon cancer cells, both tumor size and luminescent signal were assessed (Figure 2A). To confirm tumor-specific signal expression, bioluminescence intensity was compared between the tumor region and adjacent non-tumor muscle tissue, demonstrating a strong and selective signal from the tumor site (Figure 2B). We added that six mice were used for subcutaneous xenograft generation, and five out (3.0–6.0 × 10^10^ p/s/cm^2^/sr) of six tumors were selected for orthotopic implantation based on uniformity and viability, resulting in an 83.3% success rate for usable xenograft. Subsequently, tumor tissues were excised and trimmed to uniform size for transplantation into recipient mice to ensure reproducibility in the orthotopic and metastatic models (Figure 2C). The trimmed tumor fragments were weighed to verify consistency, and the variation in tissue weight was confirmed to be within 0.15 ± 0.01 mg with a coefficient of variation (CV) of 6.7%, confirming high reproducibility (Figure 2D).

### 3.2. Comparison of Orthotopic and Metastatic Colon Cancer Modeling Techniques Using Various Methods

A total of 15–21 mice (*n* = 5–7 mice per group) were used for this comparative study. To evaluate and optimize orthotopic and metastatic colon cancer modeling strategies, three different surgical approaches were compared (Figure 3A–C). In the first method, a conventional syringe injection technique (Figure 3A) was employed, in which 1 × 10^6^ bioluminescent HCT116/Luc colon cancer cells suspended in 30 μL of PBS were directly injected into the cecum. The second approach (Figure 3B) involved surgical implantation of tumor fragments, each trimmed to 0.15 mg, directly sutured to the cecal surface. The third and novel method (Figure 3C) utilized a biological bonding agent to affix uniformly sized tumor tissues onto the cecum without the need for suturing. For all models, abdominal closure was performed using the same biological adhesive.

The biological bond-based method significantly reduced operation time to under 5 min per animal, compared with 15–20 min required for the syringe injection and suturing methods. Average procedure durations were 4.8 ± 0.4 min (biological bond), 17.6 ± 1.2 min (surgical suturing), and 15.1 ± 0.8 min (syringe injection). The biological bond group showed a statically significant reduction in procedure time compared with the other groups. This offers a major advantage for high-throughput experimentation and for minimizing animal stress. To assess tumor progression and metastasis, bioluminescence imaging was conducted over a 4-week period (Figure 3D,E). In the syringe injection model, tumor signal gradually increased, but signal variability and failure of tumor establishment were observed in some animals. Tumor engraftment rates were 40% (2/5) for the syringe group, 60% (3/5) for the surgical suturing group, and 100% (5/5) for the biological boding group. Bioluminescent signal intensities (day 25) were 1.5 × 10^9^ ± 0.8 × 10^9^ (syringe injected group), 2.2 × 10^9^ ± 0.8 × 10^9^ (surgical suturing group), and 6.0 × 10^9^ ± 4.0 × 10^9^ (biological bond group) p/s/cm^2^/sr. In contrast, both the suturing and biological bond methods resulted in consistent tumor growth. However, the suturing-based model showed limited angiogenesis and a lower incidence of metastasis. The biological bond model demonstrated not only reliable tumor growth but also robust neovascularization and metastatic spread to distant organs. The incidence of liver metastasis was 0% in the injection group, 20% (1/5) in the suturing group, and 80% (4/5) in the biological group. Metastatic differences between groups were statistically significant.

These results indicate that the biological bonding technique offers a simple, time-efficient, and highly reproducible method for establishing orthotopic and metastatic colon cancer models, outperforming traditional methods in both consistency and potential.

### 3.3. Validation of Orthotopic and Metastatic Colon Cancer Modeling Using a Biological Bond-Based Implantation Technique

To validate the efficiency of the biological bond-based orthotopic and metastatic colon cancer modeling approach, in vivo and ex vivo analyses were performed following tumor implantation. Figure 4A shows bioluminescence images acquired in live animals after four weeks of monitoring. Prior to organ excision, in vivo imaging revealed that 3 out of 5 mice exhibited detectable bioluminescent signals in regions beyond the primary cecal site, suggesting potential metastasis to distant organs.

However, upon necropsy and ex vivo imaging of the liver and lungs, metastatic tumors were detected in all five animals, indicating that the biological bond-based method consistently induces metastatic progression. This result also highlights the limitation of in vivo imaging sensitivity in detecting early or low-burden metastases.

To further substantiate the metastatic consistency of the model, numerical values for bioluminescent signal intensity in each organ (cecum, liver, lung) were collected across all five animals. Although the group size (*n* = 5) is limited, signal intensity data demonstrated a relatively low inter-animal variation (mean radiance in liver: [2.0 × 10^7^] ± [1.0 × 10^7^] p/s/cm^2^/sr; lung: [0.5 × 10^7^]; cecum: [4.0 × 10^7^]). Moreover, ex vivo signals from the liver showed approximately 2-fold higher photon intensities compared with background levels, supporting the detection of metastatic lesions.

While detailed histological scoring was not performed in this study, metastases identified by bioluminescence were histologically confirmed in H&E-stained liver and lung sections (Figure 5). Future versions of this model can incorporate semi-quantitative scoring (e.g., lesion number, diameter) to better correlate imaging signals with tumor burden.

Figure 4B presents representative gross anatomical and bioluminescent images of liver tissues, confirming the presence of metastatic colon cancer lesions. Bioluminescent signals were clearly observed in liver tissues, providing direct evidence that the luminescent colon cancer successfully metastasized from the cecum to the liver.

These findings confirm that the biological bonding technique reliably supports the formation of metastatic tumors and provides a reproducible platform for modeling advanced-stage colon cancer.

### 3.4. Histological and Molecular Validation of Liver Metastasis in a Biological Bond-Based Luminescent Colon Cancer Model

To validate orthotopic and metastatic tumor formation following biological bond-based implantation of luminescent colon cancer, mice were monitored for four weeks and then sacrificed for tissue analysis. As shown in Figure 5, the cecum (colon), liver, and lungs were exercised, fixed, dehydrated, paraffin-embedded, and sectioned. Standard hematoxylin and eosin (H&E) staining was performed on tissue sections to distinguish between normal and cancerous regions. H&E staining, widely used in histopathology, enables identification of cell nuclei (stained blue by hematoxylin) and cytoplasmic structures (stained pink by eosin), allowing for the evaluation of tumor presence and pathological changes.

Results of the microscopic analysis confirmed the formation of tumor lesions in all three organs—cecum, liver, and lungs—indicating successful establishment of both orthotopic and metastatic tumors. These results clearly support the conclusion that the bioluminescent colon cancer had metastasized from the primary implantation site to distant organs, particularly the liver and lungs.

These findings collectively confirm that the biological bond-based model supports reliable hepatic metastasis of colon cancer and demonstrates the utility of both histological and molecular assays in verifying metastatic progression.

### 3.5. Evaluation of Anticancer Efficacy in a Biological Bond- and Surgery-Based Orthotopic Luminescent Colon Cancer Model

To assess the applicability of the biological bond-based orthotopic colon cancer model for anticancer drug evaluation, both biological bond and surgical implantation methods were utilized to establish luminescent HCT116/Luc colon tumors. As shown in Figure 6, bioluminescence imaging was performed the day after implantation to confirm successful model establishment.

**Figure 6 cancers-17-02266-f006:**
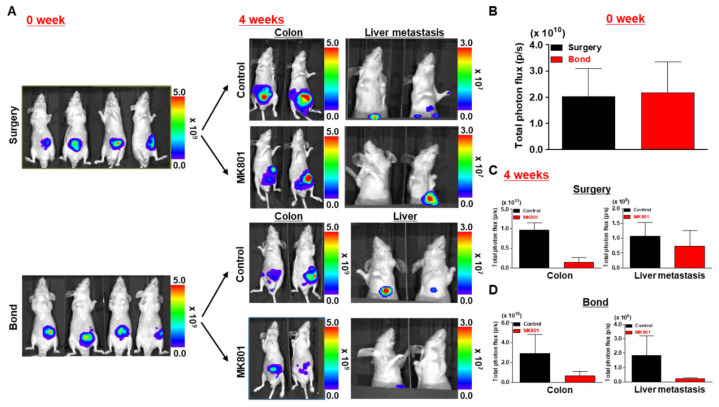
Evaluation of anticancer efficacy using biological bond- and surgery (suture)-based orthotopic colon cancer models. (**A**) In vivo bioluminescence imaging performed one day after tumor implantation confirmed successful model establishment in all groups. Mice were randomly assigned and treated with MK801 (1 mg/kg, intraperitoneally, daily on Days 1–5 post-implantation). Bioluminescent signals were monitored over 4 weeks. A reduction in tumor signal intensity was observed in both MK801-treated groups, while liver metastasis was only detected in the untreated biological bond group. (**B**) Quantification of total photon flux from the tumor regions at the final time point showed a consistent decrease in signal in the MK801-treated groups, supporting the therapeutic effect of NMDA receptor inhibition. The biological bond model also exhibited enhanced and reproducible tumor engraftment compared with the surgery-based model.

Each experimental group consisted of five animals (*n* = 5), randomly assigned to either MK801 treatment or vehicle control. MK801 was intraperitoneally administered at a dose of 1 mg/kg/day for five consecutive days (Day 1–5) post-implantation.

The anticancer agent used for validation was MK801, an NMDA receptor antagonist known to regulate tumor metabolism. Given its emerging therapeutic potential in liver cancer, MK801 was selected for this study. Mice received intraperitoneal injections of MK801 at a dose of 1 mg/kg daily from Day 1 to Day 5 post-implantation. Tumor progression and metastatic behavior were then monitored over a 4-week period using bioluminescent imaging.

At the end of the monitoring period, a reduction in tumor signal intensity was observed in both the biological bond and surgery-based groups treated with MK801, indicating therapeutic efficacy. Notably, metastatic spread to the liver was observed only in the untreated biological bond group, suggesting that the drug effectively suppressed metastasis in this model. These findings were further confirmed by quantification of bioluminescence signals using imaging analysis software 4.7x (Figure 6B), which demonstrated consistent reductions in signals in the drug-treated groups.

While detailed statistical analysis and signal quantification were not performed in this preliminary feasibility study, we acknowledge this as a limitation. Our primary objective at this stage was to test whether the biological bond-based model can feasibly support therapeutic evaluation. In future studies, we plan to include quantitative assessment of bioluminescent signal intensity, *p*-value-based comparisons, and histological validation to strengthen therapeutic claims.

Additionally, although formal toxicity evaluations were not conducted, no notable adverse effects such as weight loss or behavioral abnormalities were observed during the treatment period, suggesting preliminary tolerability of MK801 in this model.

Together, these results indicate that the biological bond-based model not only allows for reliable orthotopic tumor establishment but also promotes effective tumor engraftment and angiogenesis. Compared with conventional surgical methods, the biological bond approach offers superior suitability for evaluating the therapeutic impact of anticancer agents in orthotopic and metastatic cancer models.

## 4. Discussion

We developed and validated a novel orthotopic and metastatic colon cancer model using a biological bonding approach and demonstrated its utility in preclinical therapeutic evaluation. Our results indicate that this method provides a rapid, reproducible, and biologically relevant platform for studying tumor progression and metastasis, as well as for testing anticancer agents [35].

Traditional orthotopic xenograft models of colorectal cancer often rely on surgical implantation or direct injection of tumor cells into the cecum, which require time-consuming procedures and often result in variable engraftment efficiency and metastasis formation [36,37]. As observed in our comparative experiments, the conventional syringe injection model showed inconsistent tumor establishment and lower metastatic potential. Likewise, the suturing technique, while offering better initial tumor engraftment, was associated with limited angiogenesis, resulting in restricted metastatic spread.

In contrast, our biological bond-based model significantly reduced the implantation time to under five minutes per mouse, while also ensuring consistent tumor size, weight, and localization. This uniformity is critical for minimizing inter-animal variation and improving reproducibility in therapeutic studies. Importantly, bioluminescence imaging confirmed not only robust primary tumor growth but also metastatic dissemination to the liver and lungs, verified both by in vivo imaging and histopathological examination [38].

In our study, 3 out of 5 mice exhibited detectable metastatic signals on in vivo imaging, while all 5 mice showed clear liver and lung metastases upon ex vivo analysis, underscoring the enhanced sensitivity of direct tissue imaging. Although statistical comparisons (e.g., *p*-values) were not performed in this preliminary study, we acknowledge this limitation and plan to include detailed quantitative analyses in future studies.

Histological analyses with H&E staining clearly identified tumor lesions in the cecum, liver, and lungs, confirming successful establishment of both orthotopic and metastatic tumors [39,40]. Furthermore, molecular validation using Western blotting revealed elevated expression of colon cancer-associated markers, including HK2 and LDHA, in tumor-bearing liver tissues. These findings support the biological fidelity of our model in mimicking human colorectal cancer metastasis, particularly hepatic involvement, which is a clinically relevant hallmark of advanced-stage disease.

Although detailed histological comparisons between treated and untreated groups were not conducted, therapeutic efficacy of MK801 was observed via reduced bioluminescent signal in treated groups. Future work will incorporate immunohistochemical staining to evaluate tumor identity, proliferation, and therapy response more rigorously.

Similarly, a study by Hu et al. (2021) [35] introduced an orthotopic colon cancer model using tissue adhesive, which simplified the implantation procedure and reduced surgical burden. However, their model did not demonstrate consistent metastatic spread or allow for therapeutic evaluation, limiting its translational relevance. In contrast, our biological bond-based model not only achieves reliable engraftment and spontaneous metastasis to clinically relevant organs such as the liver and lungs but also provides a robust platform for evaluating anticancer therapies, as demonstrated by MK801 treatment in this study.

The observed superiority of the biological bond method may be attributable not only to procedural efficiency but also to enhanced adhesion and integration of tumor tissue with the host vasculature. Further mechanistic studies using markers of angiogenesis (e.g., CD31 immunostaining) are warranted to confirm this hypothesis.

It is also important to note that no signs of relapse or tumor regrowth were detected within the 4-week observation period; however, this duration may be insufficient to capture long-term tumor dynamics such as dormancy or recurrence. Longer-term follow-up studies (e.g., 8–12 weeks) are planned to better understand these aspects.

The biological bond-based model was also successfully applied in the evaluation of an anticancer agent, MK801. MK801 (Dizocilpine) is a well-known non-competitive antagonist of the N-methyl-D-aspartate (NMDA) receptor, which has been investigated for its ability to modulate tumor metabolism and inhibit cancer progression. Although it has been primarily studied in hepatocellular carcinoma, recent findings suggest that NMDA receptor subunits are also expressed in colorectal cancer cells, where they participate in glutamate-mediated signaling pathways that promote tumor growth, migration, and survival. For example, overexpression of the NMDA receptor NR1 subunit has been observed in colorectal tumor tissues and has been associated with increased proliferative capacity and poor prognosis.

Moreover, glutamatergic signaling in the tumor microenvironment has been implicated in cancer cell metabolic adaptation, angiogenesis, and metastasis. MK801’s ability to block this pathway makes it a promising agent not only for liver cancer but also for colorectal malignancies, particularly in models that aim to evaluate metabolic regulation and metastatic suppression.

In our study, MK801 was selected based on this mechanistic rationale to assess its therapeutic potential in an orthotopic and metastatic colorectal cancer model. The observed reduction in tumor progression and liver metastasis in the MK801-treated group supports the hypothesis that NMDA receptor antagonism may represent a viable therapeutic strategy in colorectal cancer.

That said, we acknowledge that off-target effects of MK801, along with the lack of systemic toxicity profiling, are limitations of the current study. Although no overt signs of toxicity were observed (e.g., behavioral changes, weight loss), future work will include formal toxicity assessment (e.g., organ histology, blood analysis).

Bioluminescence imaging revealed clear signal reduction in treated animals, especially in the biological bond group, which showed higher baseline metastatic potential. Notably, liver metastasis was observed only in the untreated biological bond group, underscoring the effectiveness of the therapeutic intervention in this context.

Taken together, these findings underscore several key advantages of the biological bonding method: (1) reduced surgical complexity and time; (2) enhanced reproducibility through standardized tumor grafting; (3) reliable formation of primary and metastatic lesions; (4) robust compatibility with noninvasive imaging and molecular validation techniques; and (5) the use of immunocompromised nude mice, although necessary for human tumor engraftment, inherently precludes evaluation of immune–tumor interactions. This limits the ability to study responses to immunotherapies or investigate the tumor microenvironment under immunocompetent conditions. Future validation in syngeneic or humanized mouse models will be essential for exploring immune responses and for testing immunotherapeutic strategies in this platform. (6) Four-week observation period employed in our current study was sufficient to monitor primary tumor growth and early metastatic spread; however, it may not fully capture late-stage metastatic progression, dormancy, or recurrence phenomena. Long-term follow-up studies (e.g., 8–12 weeks) can provide deeper insights into the metastatic timeline and therapeutic durability. (7) The clinical translatability of this model would benefit from comparative studies using clinically relevant agents, such as approved chemotherapeutics or immune checkpoint inhibitors. Additionally, incorporating co-treatment regimens with immunotherapeutics can help determine the model’s compatibility with emerging combination therapies and increase its value for preclinical screening in immune-responsive cancer types [41]. These features make the model particularly suitable for high-throughput preclinical drug screening and mechanistic studies of metastatic colon cancer.

However, some limitations should be noted. In this study, treatment groups were randomly assigned to minimize selection bias; however, blinding was not implemented during treatment administration, imaging acquisition, or histological evaluation. Although all data were analyzed using objective quantitative methods such as ROI-based IVIS imaging and Prism statistical tools, the absence of blinding may introduce potential observer bias. To further enhance the rigor and reliability of this study, future experiments will incorporate blinded assessments in addition to randomization.

While our model successfully recapitulates key features of metastatic colorectal cancer, including liver tropism, it remains a xenograft model and therefore may not fully reflect the complexity of human immune–tumor interactions [42,43,44]. Future studies incorporating syngeneic or humanized mouse systems can provide further insights into immune responses and therapeutic resistance mechanisms. Additionally, long-term monitoring beyond four weeks may be necessary to evaluate late-stage metastatic dynamics and recurrence patterns. In conclusion, we present a novel and efficient orthotopic colon cancer model using a biological bonding strategy, offering a valuable tool for cancer research. This model enables consistent tumor establishment and metastasis and proves effective for evaluating therapeutic agents such as MK801. Our approach may facilitate the development of improved cancer treatments and enhance the translational relevance of pre-clinical studies.

## 5. Conclusions

In this study, we successfully developed and validated a biological bond-based orthotopic and metastatic colon cancer mouse model using luminescent HCT116/Luc cells. This novel implantation technique demonstrated superior efficiency, consistency, and biological relevance compared with conventional syringe injection and suturing methods. The model reliably supported tumor engraftment, neovascularization, and spontaneous metastasis, particularly to the liver and lungs, closely mimicking advanced-stage human colorectal cancer.

Histological and molecular analyses confirmed the presence of metastatic lesions, and the model proved highly effective in evaluating anticancer drug efficacy, as demonstrated by the therapeutic response to MK801. Notably, the reduced procedural time and high reproducibility of the biological bonding method present clear advantages for preclinical research.

Nevertheless, certain limitations of the current study should be acknowledged. These include the use of immunodeficient mice, which precludes evaluation of immune–tumor interactions, a small sample size (n = 5 per group), a relatively short follow-up period (4 weeks), and the absence of blinding during outcome assessments.

Future studies will address these limitations by validating the model in syngeneic or humanized immune-competent mouse systems, incorporating longer-term monitoring, and expanding therapeutic testing to include immunotherapeutic agents and combination regimens. Such improvements will enhance the translational value of the model and broaden its applicability in preclinical oncological research.

Overall, this biological bond-based platform offers a powerful and scalable approach for investigating tumor biology, metastatic progression, and therapeutic interventions in colorectal cancer, providing a robust and translationally relevant platform for preclinical oncology research.

## Figures and Tables

**Figure 1 cancers-17-02266-f001:**
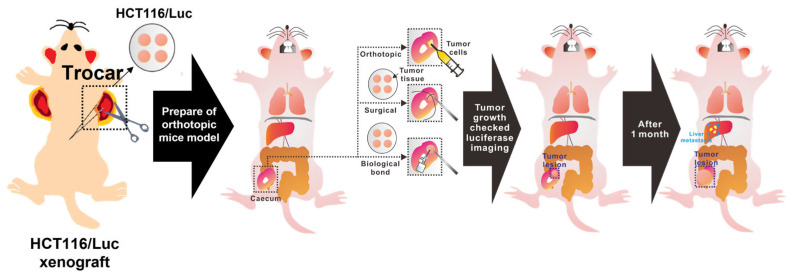
Schematic workflow for the development of a novel orthotopic and metastatic colorectal cancer model using biological tissue adhesive. The diagram illustrates: (1) subcutaneous xenograft formation using HCT116-Luc cells in nude mice; (2) harvesting and trimming of tumor tissue into standardized fragments; (3) three orthotopic implantation approaches direct injection into the cecum, surgical suturing of tumor fragments onto the cecum, and biological bonding; (4) closure of the abdominal incision using adhesive; (5) longitudinal monitoring of tumor growth and metastasis via bioluminescence imaging (IVIS); and (6) post-mortem detection of metastases in distant organs such as liver and lungs, indicated by corresponding anatomical icons.

**Figure 2 cancers-17-02266-f002:**
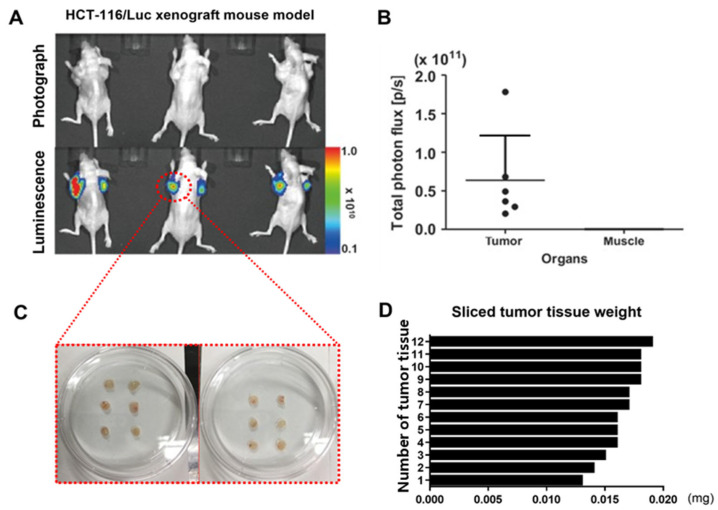
Validation and preparation of subcutaneous xenograft tissue for orthotopic transplantation in luminescent colon cancer models. (**A**) Representative photographic and bioluminescent images of nude mice bearing subcutaneous HCT116-Luc tumors 7 days post-implantation. (**B**) Quantitative comparison of bioluminescent signal intensity between tumor and surrounding normal tissue, demonstrating tumor-specific signal. (**C**) Gross morphology of excised tumor tissues, trimmed to uniform sizes for transplantation. (**D**) Histogram showing tumor fragment weight consistency (mean ± SD, n = 5), confirming reproducibility of tissue preparation for orthotopic implantation.

**Figure 3 cancers-17-02266-f003:**
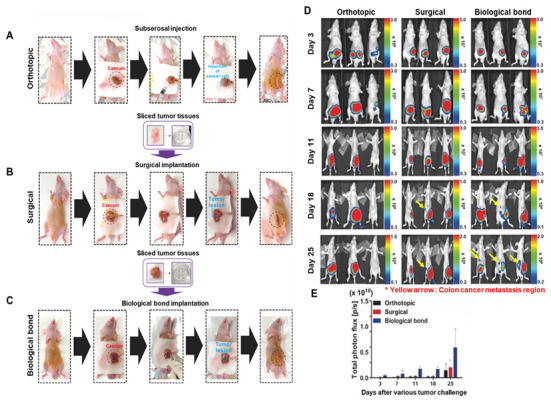
Comparative evaluation of three orthotopic colon cancer implantation techniques. (**A**) Representative schematic and bioluminescent images of the orthotopic injection model, where tumor fragments or cell suspensions are delivered via syringe to the cecum wall. (**B**) Surgical suturing model, in which tumor fragments are physically attached to the cecal surface using suture threads. (**C**) Biological bonding model, where tumor fragments are affixed to the cecum using tissue adhesive for rapid and reproducible implantation. (**D**,**E**) Longitudinal quantification of total tumor photon flux (bioluminescence signal) from Day 3 to Day 25 post-implantation. The biological bonding group demonstrates higher tumor signal intensity and lower variability over time, compared with injection and suturing groups. Asterisks indicate statistically significant differences (*p* < 0.05).

**Figure 4 cancers-17-02266-f004:**
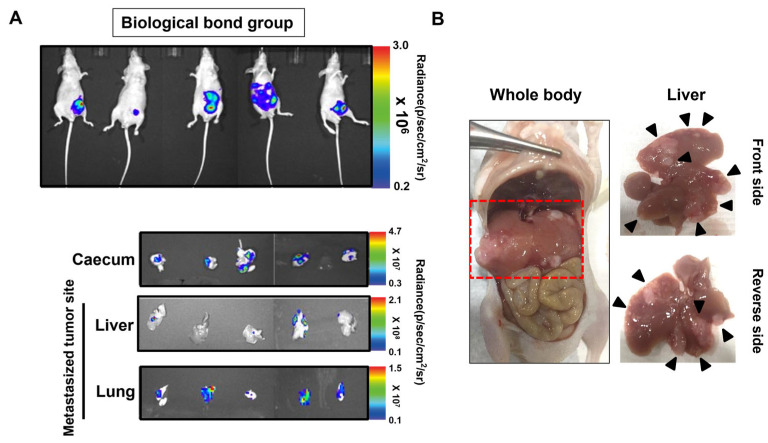
(**A**) In vivo and ex vivo bioluminescence imaging of cecum (primary tumor site), liver, and lung tissues 25 days after tumor implantation via biological bonding. In vivo images show signal from luciferase-expressing HCT116/Luc colon cancer cells. Color bars represent radiance intensity (photons/sec/cm^2^/sr), with higher values indicating stronger bioluminescent signals associated with higher tumor burden. A consistent threshold of [state threshold if known, e.g., 1 × 10^5^ p/s/cm^2^/sr] was applied to define metastatic signals. (**B**) Representative gross anatomical and ex vivo IVIS images of resected livers showing metastatic lesions on the hepatic surface, with both ventral and dorsal views presented. Bioluminescent signal in the liver confirms metastatic spread from the cecum. These results collectively demonstrate the radiance-based detection of tumor progression from luciferase-tagged colorectal cancer xenografts.

**Figure 5 cancers-17-02266-f005:**
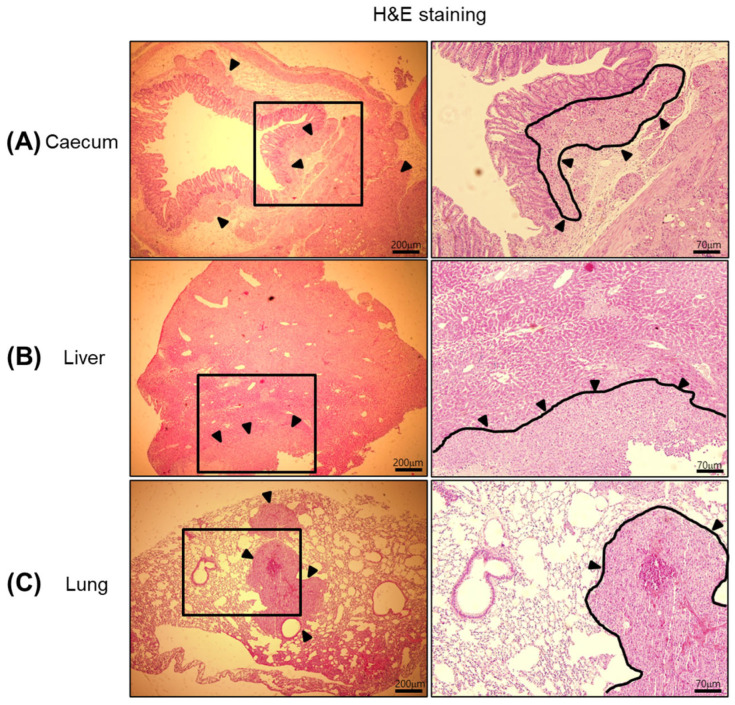
Histological validation of metastatic colon cancer in major organs following biological bond-based tumor implantation. Representative H&E-stained tissue sections from (**A**) cecum, (**B**) liver, and (**C**) lung demonstrate the presence of metastatic tumor lesions (black arrows). Metastatic foci are characterized by dense clusters of atypical epithelial cells infiltrating normal tissue architecture. These findings confirm that bioluminescent signals detected by in vivo and ex vivo IVIS imaging correspond to actual metastatic colon cancer lesions in distant organs. The results support the biological fidelity of the model in replicating advanced-stage colorectal cancer metastasis.

## Data Availability

The datasets generated and analyzed in the current study are available from the corresponding author on reasonable request.

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
