# Peer review of "Establishment of an Orthotopic and Metastatic Colorectal Cancer Mouse Model Using a Tissue Adhesive-Based Implantation Method"

_cancers, 2025, doi:10.3390/cancers17132266_

Round 1

Reviewer 1 Report

Comments and Suggestions for Authors

The authors reported a novel method to establish an orthotopic colon cancer model, using the biological bond-based tissue adhesion. They compared this new method with the conventional approaches and showed that the biological bond approach produced more metastatic tumors and responded better to drug treatment. Overall, this is an interesting study, with some convincing data. It can be improved after the authors address the following concerns:

Methods: Provide information on how tutor metastasis was detected.

Figure 2: Provide more information in the legend.

Figure 4: Have the authors detected tumor metastasis in the injection and surgical models? If they did, comparisons need to be provided.

Figure 5: Describe “arrow” and “square” in the legend.

Figure S1: The authors can make it to Figure 6. The authors could study the major pathways in colon cancer in these three models after MK801 treatment, which will help them explain the results.

Discussion: The authors may speculate why the biological bond methods showed a better response to MK801.

Author Response

Please find attached the revised manuscript and a detailed review response letter, in which we have addressed all of the reviewers’ comments point by point.

Reviewer 2 Report

Comments and Suggestions for Authors

Dear Authors, 

In the attached annotated PDF of your manuscript you may find my suggestions and recommendations related to your paper. 

Though the paper presents a very interesting topic and methodology seems to be sound and rigorous, several improvements are recommended for all the sections. 

Author Response

(The authors gave the same response as above.)

Round 2

Reviewer 1 Report

Comments and Suggestions for Authors

The authors have addressed my concerns.